# *Encephalartos natalensis*, Their Nutrient-Cycling Microbes and Enzymes: A Story of Successful Trade-Offs

**DOI:** 10.3390/plants12051034

**Published:** 2023-02-24

**Authors:** Siphelele Ndlovu, Terence N. Suinyuy, María A. Pérez-Fernández, Anathi Magadlela

**Affiliations:** 1School of Life Sciences, College of Agriculture, Engineering and Science, University of KwaZulu-Natal (Westville Campus), Private Bag X54001, Durban 4000, South Africa; 2School of Biology and Environmental Sciences, University of Mpumalanga (Mbombela Campus), Private Bag X11283, Mbombela 1200, South Africa; 3Department of Physical, Chemical and Natural Systems, Universidad Pablo de Olavide, Carretera de Utrera Km 1, 41013 Seville, Spain

**Keywords:** *Encephalartos natalensis*, microbe-symbiosis, soil nutrition, enzyme activities, soil nutrient cycling

## Abstract

*Encephalartos* spp. establish symbioses with nitrogen (N)-fixing bacteria that contribute to soil nutrition and improve plant growth. Despite the *Encephalartos* mutualistic symbioses with N-fixing bacteria, the identity of other bacteria and their contribution to soil fertility and ecosystem functioning is not well understood. Due to *Encephalartos* spp. being threatened in the wild, this limited information presents a challenge in developing comprehensive conservation and management strategies for these cycad species. Therefore, this study identified the nutrient-cycling bacteria in *Encephalartos natalensis* coralloid roots, rhizosphere, and non-rhizosphere soils. Additionally, the soil characteristics and soil enzyme activities of the rhizosphere and non-rhizosphere soils were assayed. The coralloid roots, rhizosphere, and non-rhizosphere soils of *E. natalensis* were collected from a population of >500 *E. natalensis* in a disturbed savanna woodland at Edendale in KwaZulu-Natal (South Africa) for nutrient analysis, bacterial identification, and enzyme activity assays. Nutrient-cycling bacteria such as *Lysinibacillus xylanilyticus*; *Paraburkholderia sabiae*, and *Novosphingobium barchaimii* were identified in the coralloid roots, rhizosphere, and non-rhizosphere soils of *E. natalensis*. Phosphorus (P) cycling (alkaline and acid phosphatase) and N cycling (β-(D)-Glucosaminidase and nitrate reductase) enzyme activities showed a positive correlation with soil extractable P and total N concentrations in the rhizosphere and non-rhizosphere soils of *E. natalensis*. The positive correlation between soil enzymes and soil nutrients demonstrates that the identified nutrient-cycling bacteria in *E. natalensis* coralloid roots, rhizosphere, and non-rhizosphere soils and associated enzymes assayed may contribute to soil nutrient bioavailability of *E. natalensis* plants growing in acidic and nutrient-poor savanna woodland ecosystems.

## 1. Introduction

Cycads are perennial dioecious gymnosperms regarded as “living fossils” as they possess intermediate morphological traits between angiosperms and gymnosperms [1]. Cycads originated (~265–290 Ma) in the late Palaeozoic period and were dominant during the Mesozoic era [2,3]. Cycad communities form significant vegetation in natural ecosystems as they provide critical ecosystem services such as biodiversity maintenance, carbon sequestration, and nutrient cycling [4,5,6]. Currently, cycads are distributed in the tropical and subtropical regions of America, Asia, Africa, and Oceania [7]. South Africa is classified as a centre of cycad diversity where 37 of the 66 species of the African cycad *Encephalartos* occur [3,8,9]. Cycads are regarded as the most threatened group of plants on earth, with over 60% of the 340 species threatened with a high risk of extinction [10]. Approximately 71% of South African cycad species are classified as threatened [6]. Cycads face many threats that cause a decline in their population; these threats include climate change, the presence of invasive species, their illegal harvesting for medicinal purposes, overcollection, deforestation and habitat loss, and disturbances such as fires, floods, and droughts [11,12,13]. The observed decline in cycad populations is alarming as a decline in their abundance, and probable extinction would lead to the loss of the critical ecosystem services they provide.

Cycad species grow in grasslands, sand dunes, rocky outcrops, scarp and sclerophyll forests, and areas with recurrent fires [5], and their ability to form symbioses with N-fixing, N-cycling, and P-solubilizing bacteria is likely to contribute to their ability to grow and thrive in nutrient-poor and harsh environmental conditions [5,6]. Cycads establish mutualistic associations with various microorganisms such as fungi, bacteria, and cyanobacteria in their coralloid roots [5]. Cycads are the only gymnosperms known to establish a symbiotic relationship with N-fixing cyanobacteria (*Nostoc*, *Scytonema*, and *Richelia*) and proteobacteria (*Bradyrhizobium* and *Burkholderia*) have been isolated in the coralloid roots of some cycad species [6,14,15,16]. The microbes that associate with cycads assist in plant growth by providing essential elements such as N compounds [17,18]. Additionally, soil microorganisms produce extracellular enzymes that hydrolyse and transform polymeric compounds into readily available nutrients for plant and microbe assimilation [19]. These extracellular enzymes regulate the mineralisation and cycling of terrestrial nutrients such as N, phosphate (P), and carbon (C), and these enzymes can be grouped according to their function. However, some play a role in more than one cycle [20]. Enzymes such as β-(D)-glucosaminidase and asparaginase hydrolyse chito-oligosaccharides convert asparagine into ammonia (NH_3_) and aspartic acid (C_4_H_7_NO) [21,22]. This influences N bioavailability, increasing N assimilation by plants [23]. Phosphatases hydrolyse phosphoric acid monoester into a phosphate anion [24]. Carbon cycling enzymes include dehydrogenase, β-D-cellobiohydrolase, and β-D-glucosidase. They release saccharides from glycosides and catalyse the degradation of cellotetraose and cellulose into cellobiose that will further be transformed into glucose [25]. 

South African *Encephalartos* spp. like other cycads develop coralloid roots, which host nitrogen-fixing bacteria [25] that improve soil fertility and enhance plant growth [26,27]. *Encephalartos natalensis* is widely distributed in KwaZulu-Natal savanna woodland nutrient-poor ecosystem soils, and it is near threatened in the wild [28,29]. Similar to other cycads, *E. natalensis* may thrive in nutrient-poor soils because of their symbiosis with N-fixing and other nutrient-cycling bacteria. However, the identity of the *E. natalensis*-associated symbionts with nutrient-cycling functions and their contribution to soil fertility is not well understood. Additionally, the available literature is limited to the associations of cycads with N-fixing bacteria and does not look at the associations of cycads with N-cycling and P-solubilising bacteria. Furthermore, no studies have assayed enzyme activities in *E. natalensis* rhizosphere and non-rhizosphere soils. This limited information presents a challenge in developing comprehensive conservation and management strategies for this cycad species. Moreover, elucidating *E*. *natalensis* microbe-symbiosis aligns with the KwaZulu-Natal Nature Conservation Management Act no. 9 of 1997 and the information generated from this study will feed into the knowledge of biodiversity and the enhancement of ecosystem services which aligns with South Africa’s National Biodiversity Strategy and Action Plan (NBSAP) 2015–2025. Thus, the present study identified the bacteria in *E. natalensis* coralloid roots, rhizosphere, and non-rhizosphere savanna woodland soils. The rhizosphere and non-rhizosphere soil were assayed for nutrient characteristics and soil enzyme activities to establish the relationship with the identified bacteria in the coralloid roots of *E. natalensis*. The results from this study will increase our knowledge of *E. natalensis* cycad-microbe symbiosis and its possible contribution to soil nutrient bioavailability in savanna woodland nutrient-stressed ecosystem soils. Additionally, the results of this study will help in understanding the ecosystem services of endangered cycad species with evolutionary lineages that resemble *E. natalensis* but cannot be studied due to their critical conservational status. The scientific question was the following: is the survival of the South African *E. natalensis* cycad in nutrient-deficient ecosystem soils a result of its ability to form symbiotic associations with bacteria in its coralloid roots? Hence, the aim of this research was (1) to identify bacteria in the coralloid roots, rhizosphere, and non-rhizosphere control soil of *E. natalensis* and (2) to correlate soil nutrition with the enzyme activities of the soil bacteria from the coralloid roots, rhizosphere, and non-rhizosphere control soil of *E. natalensis*. We hypothesized that the bacteria composition in the rhizosphere of coralloid roots of the South African *E. natalensis* cycad differs from that of the non-rhizosphere control soils of *E. natalensis*. Increased enzyme activities in the rhizosphere contribute to enhancing soil bioavailability that prompts *E. natalensis* persistence in the nutrient-stressed and disturbed savanna woodland ecosystem soils.

## 2. Results

### 2.1. Soil Characteristics

#### Soil Nutrients

Rhizosphere soils had a higher concentration of primary nutrients and intermediate nutrients, compared to the non-rhizosphere control soils. Micronutrient concentrations were higher in the non-rhizosphere control soil, except for Zinc which was greater in the rhizosphere soil. Rhizosphere soils showed a higher P and Magnesium (Mg) concentration than the non-rhizosphere soils, although no statistically significant differences were observed (Table 1). Nitrogen, Potassium (K), Calcium, (Ca), and Zinc (Zn) concentrations were significantly higher in the rhizosphere than in the non-rhizosphere control soils (Table 1). Rhizosphere soil pH was significantly higher (*p* < 0.05) than the pH of the non-rhizosphere control soils (Table 1). Rhizosphere soils had a higher exchange acidity than non-rhizosphere control soils, although this difference was not significant statistically (Table 1). Total cation exchange and exchange acidity were significantly higher in the rhizosphere than in the non-rhizosphere control soils (Table 1). Organic C was higher in the rhizosphere than in the non-rhizosphere control soils (Table 1). However, the rhizosphere had a lower C:N ratio than the non-rhizosphere control soils (Table 1). 

### 2.2. Bacterial Identification

#### 2.2.1. Bacterial Identification of *E. natalensis* Coralloid Roots

A total of ten bacterial strains from *E. natalensis* coralloid roots were isolated and identified (Table 2). The phosphate solubilizing bacteria isolated from the coralloid roots were *Lysinibacillus xylanilyticus* and *Paenibacillus peoriae* (Table 2). Additionally, the isolated N-fixing bacteria were *Bacillus thuringiensis*, *Bacillus pumilus*, *Bacillus safensis*, *P. peoriae*, *Paenibacillus taichungensis*, *Paenibacillus kribbensis*, *Paenibacillus taichungensis*, *Beijerinckia fluminensis*, *Lysinibacillus macrolides*, *Lysinibacillus pakistanensis*, and *Lysinibacillus xylanilyticus* (Table 2). P-solubilizing bacteria comprised 16.6% of them and the remaining 83.3% were N-fixing bacteria.

#### 2.2.2. Bacterial Identification of *E. natalensis* Rhizosphere

A total of 13 bacterial strains from *E*. *natalensis* rhizosphere were isolated and identified (Table 3). Phosphate-solubilizing bacteria such as *Caballeronia fortuita* and *Paraburkholderia steyni* were found in the rhizosphere of *E*. *natalensis* (Table 3). The identified N-fixing bacteria were *Bacillus fungorum*, *Paraburkholderia steyni*, *Paraburkholderia sabiae*, *Paraburkholderia tuberum*, *Massilia agilis*, and *Rhizobium mesosinicum* (Table 3). Furthermore, the N-cycling bacteria identified were *Caballeronia fortuita*, *P. steyni*, *Gottfriedia luciferensis*, *Bacillus pocheonensis*, *Bacillus ginsengisoli*, *Variovorax guangxiensis*, *Chitinophaga ginsengihumi*, and *Phyllobacterium brassicacearum* (Table 3). Regarding their activities, 12.5% were P-solubilizing, 37.5% were N-fixing, and the remaining 50% were in the N-cycling group.

#### 2.2.3. Bacterial Identification of *E. natalensis* Non-Rhizosphere Control Soils

A total of 10 bacterial strains from *E. natalensis* non-rhizosphere soils were isolated and identified (Table 4). The P-solubilizing bacteria included *P. steyni*, *Pseudomonas plecoglossicida*, *Novosphingobium barchaimii*, and *Methylobacterium dankookense* (Table 4). The N-fixing bacteria included *P. steyni*, *M. dankookense*, and *Olivibacter oleidegradans* were (Table 4). Additionally, *P. steyni*, *P. plecoglossicida*, *Neobacillus bataviensis*, *B. ginsengisoli*, *Sphingomonas jatrophae*, *Olivibacter jiluni*, and *Phyllobacterium brassicacearum* were identified as N-cycling bacteria (Table 4). Regarding their activities, 28.5% were P-solubilizing, 21.5% were N-fixing, and the remaining 50% were in the N-cycling group.

The rhizosphere and non-rhizosphere soils of *E. natalensis* shared the bacteria *Paraburkholderia steyni* (P-solubilizing, N-fixing, and N-cycling) and *Phyllobacterium brassicacearum* (N-cycling) (Figure 1).

### 2.3. Soil Enzyme Activities

β-(D)-Glucosaminidase and acid phosphatase enzyme activity values were similar in the rhizosphere and non-rhizosphere soils (Table 5). Nitrate reductase enzyme activity was significantly higher in the rhizosphere than in the non-rhizosphere soils (Table 5). Alkaline phosphatase activity was higher in the non-rhizosphere soils than in the rhizosphere, although the difference was not significant statistically (Table 5).

The two principal components of the soil nutrients (N and P) and their associated enzyme activities (acid phosphatase, alkaline phosphatase, β-(D)-Glucosaminidase (nmolh^−1^g^−1^), and nitrate reductase analysis explained the cumulative variability of the measured components with PCA 1 accounting for 46.6% and PCA 2 accounting for 27.0% of the total variation (Figure 2). Nitrate reductase enzyme activity was strongly correlated to the soil’s total N and β-(D)-Glucosaminidase enzyme activity was strongly correlated to the soil’s total N. Acid and alkaline phosphatase enzyme activities were strongly correlated to extractable P concentrations.

## 3. Discussion

Because cycads have coralloid roots which harbour N-fixing bacteria that convert atmospheric N to plant-usable N, it is expected that nutrient concentration in soils of *E. natalensis* rhizosphere will be higher than in non-rhizosphere control soils. The results in this study confirm that soils of *E. natalensis* rhizosphere have higher nutrient concentrations than non-rhizosphere control soils (Table 1). This suggests that *E. natalensis* contributes to or improves the soil nutrient status, a scenario that has been observed with *Cycas micronesica* [60]. With the exception of Mn which occurred in smaller concentrations, all the other nutrients in *E. natalensis* rhizosphere soils were present in higher concentrations, but with inconsistent levels of significance. For example, P and Mg concentrations in *E. natalensis* rhizosphere soils were higher than in non-rhizosphere control soils, but the difference was not significant statistically. Marler and Krishnapillai [60] and Marler and Calonje [61] showed that Mn and P were lower in soils under the canopy of *Cycas micronesica* in Guam and Tinian, respectively. The lack of significant differences in Mn and P concentration between rhizosphere and non-rhizosphere soils may be attributed to the presence of cattle grazing in the adjacent grasslands of the *E. natalensis* population in Edendale. Rayne and Aula [62] highlighted in a review that the application of cattle manure led to an increase in Mn and P concentrations in the soil. Additionally, the high amounts of other nutrients in the non-rhizosphere control soils may be attributed to cattle manure as suggested by Rayne and Aula [62].

*Encephalartos natalensis*-microbe symbionts and associated extracellular enzymes in soils are essential to enhancing soil nutrient inputs in savanna woodland disturbed ecosystem soils. Soil microbes are critical contributors to the mineralization and cycling of major soil nutrients and play an important role in promoting plant growth and development in natural ecosystems [56]. The most abundant group of microorganisms that occur in soils is bacteria and some of these culturable bacteria include the following genera: *Klebsiella*, *Paenibacillus*, *Lysinibacillus*, *Bacillus*, *Pseudomonas*, *Bradyrhizobium*, *Rhizobium*, and *Enterobacter* [63]. Bacteria play an important role in soil nutrient recycling, soil structure maintenance, and plant growth promotion [52,60,61]. Studies have demonstrated that indigenous growth-promoting bacteria contribute to P solubilization and N fixation, thus making P and N bioavailable for plant uptake in nutrient-deprived ecosystem soils [62]. The coralloid roots of *E. natalensis* predominated N-fixing bacteria (83.3%) indicate that the lack of N in the soil (Table 1) triggers biological N fixation to ensure plant uptake and survival. As P is needed in the BNF, and its concentrations in the soils are scarce, plants promote simultaneous symbiosis with the P-solubilizing bacteria. This way, plants are self-sufficient in terms of the deficient essential elements for plant growth (N and P). The immediate consequence is an increase of N in the rhizosphere and concomitantly, the increase of N-cycling bacteria that mobilize this nutrient. This explains the increased proportion of N-cycling bacteria both in the rhizosphere and non-rhizosphere soils (50% in both cases). In addition to that, bacteria in the *Bacillus*, *Pseudomonas*, *Paenibacillus*, and *Lysinibacillus* genera are known as plant growth-promoting rhizobacteria and for enhancing plant performance. *Bacillus* spp. secrete phytohormones such as (Indole-3-acetic acid) IAA, (Gibberellic acid) GA3, and kinetin and upregulate chlorophyll synthesis [29,64,65]. *Lysinibacillus macrolides* are N-fixing bacteria that enhance the total N content of soils and decompose organic matter [66]. The *Paenibacillus polymyxa* strain ZM27 has Zn solubilizing properties, produces exopolysaccharides, indoles acetic acid, solubilizes phosphate, produces siderophores, and 1-aminocyclopropane-1-carboxylate ACC-deaminase catalase activity [67]. *Beijerinckia fluminensis* is a multifarious plant growth-promoting bacteria that produces exopolysaccharides (EPS), hydrogen cyanide (HCN), extracellular enzymes, indole-3-acetic acid, ammonia, siderophore, ACC-deaminase and plays a role in P solubilization [68]. *Paraburkholderia* is a bacterial genus that usually occurs in low-pH agricultural and forest soil environments and contributes to several ecological processes, which include N_2_ fixation, mineral weathering, and decomposition of plant litter (both lignin and cellulose) [69,70,71,72]. The *Massilia* bacteria are known to synthesize enzymes and various secondary metabolites and play a role in P dissolution [48]. *Paraburkholderia*, *Pseudomonas*, *Novosphingobium*, and *Methylobacterium* species play a role in P solubilization [73,74], producing phosphatase enzymes [49] and producing hormones promoting plant growth [7,75,76]. *Encephalartos natalensis* occurs in savannah and grassland ecosystems which are characterized by nutrient-poor and acidic soils [77,78]. *Encephalartos natalensis* rhizosphere and non-rhizosphere soils richness in plant growth-promoting bacteria can be interpreted as one of the reasons why the species is widely distributed in nutrient-deficient soils. The presence of these different bacterial strains in *E. natalensis* rhizosphere and non-rhizosphere control soils indicates that microbiota confers the plant nutritional advantages to cope with the impoverished soils as the bacteria play a crucial role in nutrient cycling, increasing the bioavailability of soil nutrition for *E. natalensis* and surrounding plants growing in savanna woodland ecosystems.

Soil extracellular enzymes are important bioindicators of soil microflora metabolism and they provide information about soil quality, soil fertility, and the productivity status of soils [79]. Soil extracellular enzymes play a significant role in the conservation and recycling of key nutrients in nutrient-limited ecosystem soils [80]. Soil extracellular enzymes mineralize and recycle nutrients such as N, P, and C in soil increasing the bioavailability of the nutrients for uptake by plants [81]. Acetylglucosaminase, urease, and β-D-Glucosaminidase are enzymes involved in N cycling [14,15,82]. However, β-D-Glucosaminidase has also been reported to play a role in C-cycling and plays a significant role in the biological control of plant pathogens [82,83]. Phosphatases are responsible for P mineralization and cycling in soils [84]. Additionally, phosphatases produce P by hydrolysing phosphoric acid monoester to phosphate anions [85]. Acid and alkaline phosphatase enzymes are some examples of enzymes that have been studied extensively and their activity is strongly influenced by P availability and soil pH [86,87]. Magadlela et al. [88] observed increased β-Glucosaminidase, acid, and alkaline phosphatases activity in P-deficient and acidic soils compared with P-rich soils. These enzymes are linked to the bioavailability of nutrients in these nutrient-limited ecosystems through cycling N and P [89,90]. Additionally, Gavrilova et al. [91] and Speir et al. [92] suggested that increased enzyme activities in soils are regulated by soil P and N deficiency. Alkaline and acid phosphatase activities were positively correlated to the extractable P concentrations in the rhizosphere and the non-rhizosphere soils of *E. natalensis* widely distributed in the disturbed savanna woodland ecosystem at Edendale. Gavrilova et al. [91] and Speir et al. [92] found that the P concentration of soils is directly proportional to phosphatase activities. This positive correlation between acid/alkaline phosphatase and extractable soil P concentration highlights that the bioavailability of soil P is positively correlated to the production of phosphatases in soils and that phosphatases contribute to soil P nutrition in this cycad’s savanna woodland ecosystem soils. A study conducted by Kitayama [93] showed that soil phosphatase activities were negatively correlated with the pool size of soil organic P fractions, suggesting that the bioavailability of P determines the activity of phosphatases. Furthermore, Kitayama [93] reported that soil microbes increase phosphatases exudation in response to the chronic shortage of soil P, highlighting the functional role of microbes and associated enzymes in the efficient solubilization of P. The results of the PCA revealed a positive correlation between the β-(D)-Glucosaminidase activities and total N concentrations in both the rhizosphere and non-rhizosphere soils. Additionally, the nitrate reductase and total N concentrations in the rhizosphere and non-rhizosphere soils were positively correlated. β-(D)-Glucosaminidase is an enzyme that catalyzes the hydrolysis of chitin [94]. The hydrolysis of chitin is a crucial step in the cycling of soil carbon (C) and nitrogen (N). Once chitin is hydrolysed, it is converted into amino sugars [84]. These amino sugars are important sources of mineralizable N in soils [84]. Previous studies conducted by Cenini et al. [94] and Ekenler et al. [95] suggested that β-(D)-Glucosaminidase activities are positively related to soil N concentrations, these results correlate with our findings. Previous studies by Ekenler and Tabatabai [96] evaluated the relationship between N mineralization indexes and β-(D)-Glucosaminidase across 6 agroecological zones in the Northern Central region of the United States of America, and the results of the study revealed a significant correlation between β-(D)-Glucosaminidase activity and total N [96]. Cheng et al. [97] found that soil nitrate reductase was positively correlated with soil N concentration, highlighting the contribution of the enzyme nitrate reductase to soil N nutrition. Therefore, β-(D)-Glucosaminidase and nitrate reductase play a significant role in N mineralization in the disturbed savanna woodland ecosystem at Edendale.

## 4. Materials and Methods

### 4.1. Study Sites and Target Species

This study was conducted in a disturbed savanna woodland ecosystem at Edendale, Pietermaritzburg in the KwaZulu-Natal (KZN) province of South Africa where the target species *E. natalensis* is widely distributed. The soils at Edendale have a high clay content. *Encephalartos natalensis* rhizosphere and non-rhizosphere soils were collected from one large population (n > 500) at Edendale. The sampled plant coordinates are not provided due to its red data listing [20] and conservation concerns as stated by sample collection permits issued by EzeMvelo KZN wildlife guided by The National Environmental Management Act (NEMA), Act 107 of 1998 and its amendments: National Environmental Management: Biodiversity Act (NEMBA), Act 10 of 2004, its amendments and regulations including the Threatened or Protected Species (TOPS) regulations of 2007. The Edendale savanna woodland is prone to overharvesting of *E. natalensis* bark for muthi/traditional medicine, frequent fires from surrounding grassland, overgrazing by cattle, and it is dominated by invasive plants, such as, *Lantana camara* and *Isoglossa woodii*. Also, some *E. natalensis* plants are strangled by strangler figs (*Ficus* spp).

### 4.2. Soil Sampling and Soil Nutrition Analysis

Twenty mature individual *E. natalensis* plants were selected for the study. The sampled plants were between 15–30 m apart from each other. Soil samples were collected from 0–10 and 10–20 cm depths underneath the canopy of the cycad trees at a maximum distance of 1 m from the plant. The 0–10 and 10–20 cm depths were estimated with a measuring tape. This depth is considered the portion of soil in closer contact with roots and where maximum microbial activity is expected. The rhizosphere soils were collected at the four cardinal points at distances of about 30 cm from the stem and at the leaf canopy drip line. Similarly, the non-rhizosphere control soils (similar depths) were collected from non-target sites defined by a radius of five meters from the base of each target plant as control. The direction of the control site was randomly selected using the cardinal points North, East, South, and West of the target plant. The collected soil in each point (10 sub-points) was transferred to a bucket and thoroughly mixed. In total, 10 compound samples were collected per site. A portion of each compound soil sample was stored in sterile plastic bags in a refrigerator at 4 °C until chemical and biological analyses were conducted. For total nutrient analysis, the soil samples were air-dried, sieved to less than 2 mm, and 50 g of each with five replicates were sent for total P, N, K, pH, acidity exchange and total cation analysis at the KwaZulu-Natal Department of Agriculture and Rural Development’s Analytical Services Unit, Cedara, South Africa. Ground soil samples were analyzed for soil total N with the Automated Dumas dry combustion method using a LECO CNS 2000 (Leco Corporation, MI, Detroit, USA) and pH (using a KCl solution). Soil ambic-2-extractable phosphorus and K in the soil samples were measured using the atomic absorption method. This involved the extraction of a 2.5 mL soil solution with a 25 mL ambic-2 solution at a pH of 8. The mixture was stirred at 400 rpm for 10 min using a multiple stirrer and filtered using Whatman No. 1 paper. Refer to Manson and Roberts [30] for the detailed methodologies. An additional five soil subsamples (250–300 g) from each treatment were used for microbial identification and enzymatic analysis.

### 4.3. Soil Serial Dilutions and Bacterial Extraction

Three-folds of soil serial dilutions were conducted for each 10 g soil sample, and 100 μL of each dilution was used to inoculate nutrient agar plates. Phosphate solubilizing bacteria were isolated and grown on Pikovskaya’s agar plates which contained tricalcium phosphate (TCP) as the P source. The N-cycling bacteria were grown on Simmons citrate agar plates which contained citrate as a carbon (C) source and inorganic ammonium salts as the only source of N and the N-fixing bacteria were grown on the Jensen’s media agar (N-free media) [20]. Each selective media plate was replicated three times and incubated for 5–14 days at 30 °C. Pure colonies were obtained by repeated streaking onto fresh culture plates [20].

### 4.4. Coralloid Roots Surface Sterilization and Bacterial Extraction

Four lumps of coralloid roots (50 mm diameter) visible above-ground forming a big mask were collected from each randomly selected mature plant of *E. natalensis* in the Edendale population (n > 500). The coralloid roots were rinsed with distilled water and stored in ice. At the laboratory, 70% (*v*/*v*) ethanol was used to sterilize the coralloid roots for 30 s, the coralloid roots were then treated with 3.5% (*v*/*v*) sodium hypochlorite solution for 3 min. Thereafter, using distilled water, the coralloid roots were rinsed 10 times and stored in airtight vials that contained cotton wool and silica gel, then stored at 4 °C. Protocols as per Magadlela et al. [28] and Matiwane et al. [29] were used to conduct the bacterial isolation. For the bacterial extraction, the coralloid roots were subjected to 15% glycerol; thereafter, they were crushed using sterile tips. Once extracted, the bacteria were grown in sterile Petri dishes as before. Phosphate solubilizing bacteria were isolated and grown on Pikovskaya’s agar plates which contained tricalcium phosphate (TCP) as the P source. The N-cycling bacteria were grown on Simmons citrate agar plates which contained citrate as a C source and inorganic ammonium salts as the only source of N, and the N-fixing bacteria were grown on the Jensen’s media agar (N-free media). Each selective media plate was replicated three times and incubated for 5 days at 30 °C. Pure colonies were obtained by repeated streaking [20].

### 4.5. Coralloid Roots, Rhizosphere and Non-Rhizosphere Soils Bacterial Amplification, Sequencing, and Identification

Small portions of the 16S rDNA genes of pure bacterial colonies extracted from the coralloid roots, rhizosphere and non-rhizosphere soils were amplified using polymerase chain reaction (PCR). The specific primers used were 63F (5′-CAG GCCTAACACATGCAAGTC-3′) and 1387R (5′-GGGCGGTGTGTACAA GGC-3′) as per Magadlela et al. [28]. The PCR conditions were as follows: Initial denaturation was carried out at 94 °C for 5 min, 30 cycles of denaturation at 94 °C for 30 s; annealing was carried out at 55 °C for 30 s; final elongation was carried out at 72 °C for 10 min. Each 25 µL PCR reaction included the following: 11 µL sterile distilled water, 1 µL bacterial colony, 0.25 µL of the forward primer, 0.25 µL of the reverse primer, and 12.5 μL Emerald AMP master mix (Takara Bio supplied by Separations, South Africa). Thereafter, the results were viewed using agarose gel electrophoresis prepared using TAE buffer (1%). The PCR products were sent for sequencing at Inqaba Biotechnical Industries (Pty) Ltd., Pretoria, South Africa. The resulting sequences were edited and subjected to BLASTN searches for bacterial identification (National Centre for Biotechnology Information, NCBI). https//:www.ncbi.nlm.nih.gov (accessed on 31 August 2022).

### 4.6. Soil Enzymatic Studies

Nitrogen-cycling and P-cycling enzyme activities (beta- glucosaminidase, acid phosphatase, and alkaline phosphatase) were conducted using the florescence-based method adapted from Jackson et al. [33] and this was expressed in nmolh^−1^g^−1^. Five grams of each soil sample was homogenized at low speed in 50 mL ultrapure H_2_O at 4 °C for 2 h. The resulting supernatants were transferred into 96-well microplates, thereafter the 4-MUB-phosphate substrate was added for P-cycling enzymes and 4-MUB-N-acetyl-β-D-glucosaminide was added for the N-cycling enzymes. Sample runs consisted of 200 µL of soil samples of soil aliquots plus 50 µL of the substrate. Samples, standards (200 µL buffer + 50 µL standard), quench standards (200 µL soil aliquots + 50 µL substrate), and blanks (250 µL buffer) were incubated for 2 h at 30 °C using 0.5 M of NaOH to stop the reaction. The fluorescent absorbance was measured at 450 nm using an Apex Scientific microplate reader (Durban, South Africa). It is important to note that before determining acid phosphate activity, both the standards and buffer must have a pH of 5.

Nitrate reductase activities were measured with an adapted method of Bruckner et al. [34]. Five grams of each soil sample was transferred in a solution containing 4 mL of 0.9 mM 2.4-dinitrophenol, 1 mL of 25 mM KNO_3_, and 5 mL of ultrapure H_2_O in a sealed centrifuge tube (50 mL). The mixture was vigorously mixed before being incubated in the dark for 24-h at a temperature of 30 °C. After incubation, 10 mL of 4 M KCl solution was added to each sample and briefly mixed. Thereafter they were passed through Whatman number 1 filter paper. The enzymatic reaction was initiated by adding 2 mL of the filtrate to 1.2 mL of 0.19 M ammonium chloride buffer (pH 8.5) and 800 µL of the colour reagent (1% sulphanilamide in 1 N HCl and 0.2% N-(1-naphthyl) ethylenediamine dihydrochloride (NEDD) before incubation for 30 min in the dark at 30 °C. The absorbance was measured at 520 nm using an Agilent Cary 60 UV-Vis spectrophotometer (Agilent, Santa Clara, CA, USA). The amount of nitrite (NO_2_) released into the medium was expressed as 0.1 µmolh^−1^g^−1^.

### 4.7. Statistical Analysis

R studio (R version 4.2.0) was used for all analyses. An independent sample *t*-test was used to determine whether there is a significant difference between the soil characteristics and soil enzymatic activity of *Encephalartos natalensis* rhizosphere and non-rhizosphere soils. The assumption of normal distribution of data was tested using a 1-Kolmogorov-Smirnov test and the assumption of homogeneity of variance was tested using Levene’s test. Where these assumptions were not met, a non-parametric (Wilcoxon signed-rank test) alternative was used. The independent sample *t*-test procedure was performed using the statistical package car, function LeveneTest of R (4.2.0). Relations between the soil nutrient concentrations (total N and extractable P) and their associated enzyme activities of *E. natalensis* rhizosphere and non-rhizosphere soils were determined using principal component analysis (PCA). Principal component analysis procedures were performed using R (4.2.0) using statistical package gg plot, function pr comp.

## 5. Conclusions

*Encephalartos natalensis* species’ wide distribution in nutrient-poor and disturbed ecosystems such as acidic and nutrient-poor savanna woodland ecosystems may be linked to their established symbioses with N-fixing bacteria that enrich the rhizosphere and surrounding soils in nutrients. These nutrients favour the presence of nutrient-mineralizing microbes and associated extracellular enzymes in these ecosystems. Microbes identified in coralloid roots, rhizosphere, and non-rhizosphere soils and associated extracellular enzymes may contribute to soil nutrient inputs in savanna-woodland ecosystems.

## Figures and Tables

**Figure 1 plants-12-01034-f001:**
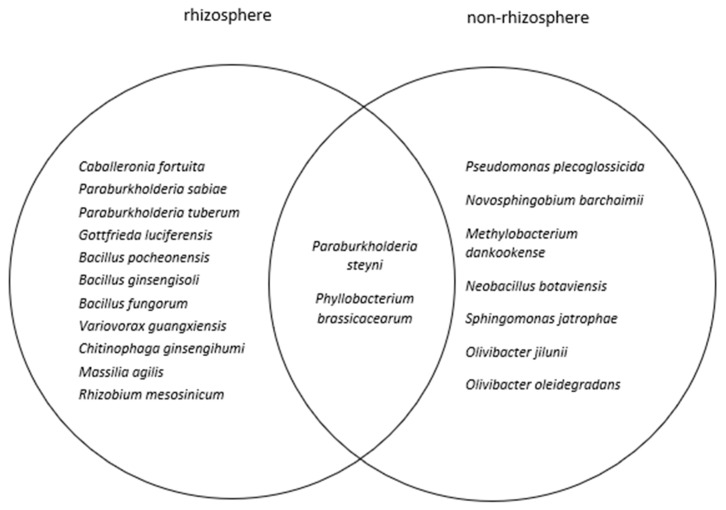
Venn diagram of bacteria shared between the rhizosphere and non-rhizosphere soils of *E. natalensis*.

**Figure 2 plants-12-01034-f002:**
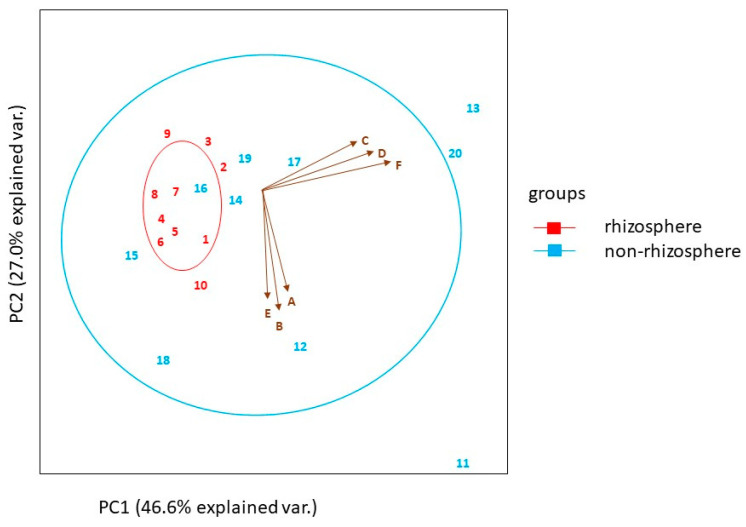
Correlation between the soil nutrients and associated enzyme activities of *E. natalensis* rhizosphere and surrounding soils. Soil characteristics are represented as follows: A = nitrate reductase (µmolh^−1^g^−1^), B = total nitrogen concentration (mg.kg^−1^), C = acid phosphatase (nmolh^−1^g^−1^), D = extractable phosphorus concentration (mg.kg^−1^), E = β-(D)-Glucosaminidase (nmolh^−1^g^−1^), F= alkaline phosphatase (nmolh^−1^g^−1^). Principal component analysis (PCA).

**Table 1 plants-12-01034-t001:** Soil characteristics of *Encephalartos natalensis* rhizosphere and non-rhizosphere soils collected in a disturbed savanna woodland ecosystem at Edendale, Pietermaritzburg in the KwaZulu-Natal (KZN) province of South Africa. The results for primary nutrients, intermediate nutrients, micronutrients, and soil relative acidity are represented as mean ± SE. Differing letters show significant differences (independent sample *t*-test, *p* ≤ 0.05, *n* = 20).

	Rhizosphere	Non-Rhizosphere
Primary nutrients (mg.kg^−1^)		
Total nitrogen	4187.01 ± 567.76 ^a^	3223.90 ± 487.21 ^b^
Extractable phosphorus	11.83 ± 3.66 ^a^	9.24 ± 0.94 ^a^
Extractable potassium	537.76 ± 50.27 ^a^	201.15 ± 56.66 ^b^
Extractable intermediate nutrients (mg.kg^−1^)		
Magnesium	620.96 ± 44.48 ^a^	555.75 ± 85.87 ^a^
Calcium	4644.63 ± 829.93 ^a^	3542.67 ± 1121.89 ^b^
Extractable micronutrients (mg.kg^−1^)		
Zinc	5.54 ± 0.54 ^a^	3.69 ± 0.57 ^b^
Manganese	50.84 ± 4.72 ^a^	75.94 ± 7.82 ^a^
Copper	74.83 ± 37.36 ^a^	89.77 ± 27.50 ^b^
Soil relative acidity		
pH	5.75 ± 0.71 ^a^	5.14 ± 0.42 ^b^
Exchange acidity (cmolc.kg^−1^)	0.08 ± 0.07 ^a^	0.05 ± 0.02 ^a^
Total cation exchange (cmolc.kg^−1^)	29.61 ± 4.62 ^a^	22.62 ± 6.37 ^b^
Parameter		
Organic Carbon (%)	5.24	4.21

**Table 2 plants-12-01034-t002:** Molecular identification of the bacterial community isolated from the coralloid roots of *E. natalensis* growing in a disturbed savanna woodland ecosystem at Edendale, Pietermaritzburg in the KwaZulu-Natal (KZN) province of South Africa.

Family	Scientific Name	Accession Number	Similarity (%)	Function
Bacillaceae	*Lysinibacillus xylanilyticus*	NR_116698.1	99.93	P solubilizing [30] N-fixing [30]
	*Lysinibacillus macrolides*	NR_114920.1	99.06	N-fixing [31]
	*Lysinibacillus pakistanensis*	NR_113166.1	99.02	N-fixing [32]
	*Bacillus thuringiensis*	MG470721.1	99.72	N-fixing [33]
	*Bacillus pumilus*	MN581190.1	99.93	N-fixing [34]
	*Bacillus safensis*	CPO43404.1	99.58	N-fixing [35]
Paenibacillaceae	*Paenibacillus peoriae*	NR_117742.1	100	P solubilizing [36] N-fixing [37]
	*Paenibacillus taichungensis*	NR_044428.1	96.70	N-fixing [38]
	*Paenibacillus kribbensis*	NR_025169.1	99.04	N-fixing [38]
Beijerinckiaceae	*Beijerinckia fluminensis*	NR_116306.1	99.76	N-fixing [39]

**Table 3 plants-12-01034-t003:** Molecular identification of the bacterial community isolated from the *E. natalensis* rhizosphere soils growing in a disturbed savanna woodland ecosystem at Edendale, Pietermaritzburg in the KwaZulu-Natal (KZN) province of South Africa.

Family	Scientific Name	Accession Number	Similarity (%)	Function
Burkholderiales	*Caballeronia fortuita*	NR_145600.1	99.09	P solubilizing [40] N cycling [41]
	*Paraburkholderia steyni*	NR_164972.1	99.19	P solubilizing [42] N-fixing [43] N cycling [43]
	*Paraburkholderia sabiae*	NR_115261.1	99.28	N-fixing [44]
	*Paraburkholderia tuberum*	NR_118081.1	99.83	N-fixing [45]
Bacillaceae	*Gottfrieda luciferensis*	NR_025511.1	98.41	N cycling [46]
	*Bacillus pocheonensis*	NR_041377.1	99.62	N cycling [46]
	*Bacillus ginsengisoli*	NR_109068.1	98.93	N cycling [47]
	*Bacillus fungorum*	NR_170494.1	90.10	N-fixing [47]
Comamonadaceae	*Variovorax guangxiensis*	NR_134828.1	99.30	N cycling [48]
Chitinophagaceae	*Chitinophaga ginsengihumi*	NR_134000.1	99.19	N cycling [49]
Phyllobacteriaceae	*Phyllobacterium brassicacearum*	NR_043190.1	91.3	N cycling [50]
Oxalobacteraceae	*Massilia agilis*	NR_157770.1	98.51	N-fixing [51]
Rhizobiaceae	*Rhizobium mesosinicum*	NR_043548.1	99.26	N-fixing [52]

**Table 4 plants-12-01034-t004:** Molecular identification of the bacterial community isolated from the non-rhizosphere soils of *E. natalensis* growing in a disturbed savanna woodland ecosystem at Edendale, Pietermaritzburg in the KwaZulu-Natal (KZN) province of South Africa.

Family	Scientific Name	Accession Number	Similarity (%)	Function
Burkholderiales	*Paraburkholderia steyni*	NR_164972.1	98.41	P solubilizing [45] N-fixing [43] N cycling [53]
Pseudomonadaceae	*Pseudomonas plecoglossicida*	NR_114226.1	98.38	P solubilizing [44] N cycling [45]
Erythrobacteraceae	*Novosphingobium barchaimii*	NR_118314.1	99.65	P solubilizing [54]
Methylobacteriaceae	*Methylobacterium dankookense*	NR_116545.1	99.77	P solubilizing [55] N-fixing [56]
Bacillaceae	*Neobacillus bataviensis*	NR_114093.1	99.51	N cycling [57]
	*Bacillus ginsengisoli*	NR_109068.1	83.02	N cycling [58]
Sphingomonadaceae	*Sphingomonas jatrophae*	NR_159248.1	85.41	N cycling [58]
Sphingobacteriaceae	*Olivibacter jilunii*	NR_109321.1	99.30	N cycling [59]
	*Olivibacter oleidegradans*	NR_108900.1	98.41	N-fixing [59]
Phyllobacteriaceae	*Phyllobacterium brassicacearum*	NR_043190.1	90.30	N cycling [59]

**Table 5 plants-12-01034-t005:** Soil enzyme activities in the rhizosphere and non-rhizosphere soils of *Encephalartos natalensis* growing in a disturbed savanna woodland ecosystem at Edendale, Pietermaritzburg in the KwaZulu-Natal (KZN) province of South Africa. Results are represented as mean ± SE. Differing letters show significant differences (independent sample *t*-test, *p* ≤ 0.05, *n* = 20).

Enzyme Activity	Rhizosphere	Non-Rhizosphere Soils
β-(D)-Glucosaminidase (nmolh^−1^g^−1^)	20.47 ± 0.95 ^a^	20.44 ± 1.03 ^a^
Nitrate reductase (µmolh^−1^g^−1^)	4496 ± 2116.6 ^a^	3178.87 ± 1408.48 ^a^
Acid phosphatase (nmolh^−1^g^−1^)	13.13 ± 2.03 ^a^	13.50 ± 3.98 ^a^
Alkaline phosphatase (nmolh^−1^g^−1^)	15.88 ± 4.10 ^a^	19.47 ± 6.81 ^a^

## Data Availability

All raw data and R scripts will be available upon request from Anathi Magadlela, email: MagadlelaA@ukzn.ac.za.

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
