# Peer review of "Encephalartos natalensis*, Their Nutrient-Cycling Microbes and Enzymes: A Story of Successful Trade-Offs"

_plants, 2023, doi:10.3390/plants12051034_

Round 1
Reviewer 1 Report
The paper describes how cycad cultivation on soil nutrient content (Organic C, Organic N, N, P, K, Mg, Ca, Zn, Cu, and Mn), enzyme activity (Glucosaminidase, Nitrate reductase, Acid phosphatase, and Alkaline phosphatase), and culturable N and P cycling related bacterial species by comparing the rhizosphere and non-rhizosphere. The topic is worth exploring and appropriate for plants. Overall, the manuscript is well-written. However, I do have two major concerns:
(1) In the title, it is stated that “the natal giant cycad associated nutrient-cycling microbes and enzymes contribute to soil nutrient inputs…..”. However, data in the current Table 1, Table 5, and Fig 1 is far from sufficient to allow us to draw this conclusion. In table 1, although the differences in some nutrients (N, K, Zn, and Cu) between rhizosphere and non-rhizosphere are significant, the differences in other nutrients (P, Mg, and Mn) between rhizosphere and non-rhizosphere are not significant. In table 5, the differences in all four enzymes between rhizosphere and non-rhizosphere are not significant. In Fig 1, the correlations in Figs 1A, 1B, 1E, 1F, and 1G are not significant (p >0.05).
It might be helpful to conduct a one-way ANOVA (besides the existing post hoc test) to check the main effect of soil type/plant cultivation on all tested variables (nutrient contents, enzyme activities) and conduct a correlation analysis to see the relationships between nutrient contents and enzyme activities. Instead of presenting correlations in the two soils separately as shown in Fig 1, all data should be analyzed together because non-rhizosphere and rhizosphere are just two levels of the same factor (soil type/plant cultivation). Non-rhizosphere should be treated as the control for comparison. By adding an ANOVA table and a correlation table, the authors would be able to see whether plant cultivation impacts a certain nutrient or enzyme activity or not and whether there is any interesting relationship between nutrient content and enzyme activity.
(2) In this study, considerable efforts have been made to identify the N-cycling microbes and P-solubilizing microbes in the roots, rhizosphere, and non-rhizosphere; however, the data presentation is quite descriptive and lacks meaningful insights. Although <10% (or even <1%) of all microbes are culturable in the lab, comparing the unique and shared microbes between rhizosphere and non-rhizosphere should still be able to give us at least some insights about who are and who are not there and who might be key players in N, P cycling. To do this, the authors can consider adding a three-way Venn Diagram to see what microbes are unique and what microbes are shared.
I am not sure whether the authors did sequencing for the rhizosphere soil and non-rhizosphere soil, respectively, as well. The soil sequence data would allow the authors to compare the “population” of the microbes between the rhizosphere soil and the non-rhizosphere soil. Even though the two soils share the same microbial “species”, the “populations” in the two soils could still have a significant difference.
Please see more of my comments below:
1. In Table 1, please clarify what Nitrogen content it is. Is it NH4-N, NO3-N, inorganic N, or total N?
2. Lines 177-178, 180-181: the statements are not correct/precise because the p values in Figs 1A, 1B, 1E, and 1F are >0.05.
3. Lines 188-198: keep the unit formats consistent. Either use “-1” or “/” but not both.
4. Discussion Lines 200-213: keep the font, size, and color consistent.
5. Author's name is needed for some citations in the text. For example, in line 216 “[46] and [47] described …..” should be replaced with “ Adeniyan et al. [46] and Akinrinlola et al. [47] described….” Double-check lines 255, 258, 262, 267, 269, 274, 277, and 288.
6. Lines 300-301 “Soil samples were collected from 0-10 and 10-20 cm depths ….”. Please clarify what soil (0-10 cm, 10-20 cm, or both) was used for soil nutrient analyses and enzyme activity assays.
7. In 4.1.2, the methods and references for soil nutrient analysis are missing. Please clarify those.
8. In 4.1.3 and 4.1.4, the DNA extraction method and reference are missing. Please clarify those.
9. In 4.1.3, the reference for the agars is missing. Please clarify that.
Author Response
Reviewer 1:
The paper describes how cycad cultivation on soil nutrient content (Organic C, Organic N, N, P, K, Mg, Ca, Zn, Cu, and Mn), enzyme activity (Glucosaminidase, Nitrate reductase, Acid phosphatase, and Alkaline phosphatase), and culturable N and P cycling related bacterial species by comparing the rhizosphere and non-rhizosphere. The topic is worth exploring and appropriate for plants. Overall, the manuscript is well-written. However, I do have two major concerns:
In the title, it is stated that “the natal giant cycad associated nutrient-cycling microbes and enzymes contribute to soil nutrient inputs…..”. However, data in the current Table 1, Table 5, and Fig 1 is far from sufficient to allow us to draw this conclusion. In table 1, although the differences in some nutrients (N, K, Zn, and Cu) between rhizosphere and non-rhizosphere are significant, the differences in other nutrients (P, Mg, and Mn) between rhizosphere and non-rhizosphere are not significant. In table 5, the differences in all four enzymes between rhizosphere and non-rhizosphere are not significant. In Fig 1, the correlations in Figs 1A, 1B, 1E, 1F, and 1G are not significant (p >0.05).
Reviewer comment:
It might be helpful to conduct a one-way ANOVA (besides the existing post hoc test) to check the main effect of soil type/plant cultivation on all tested variables (nutrient contents, enzyme activities) and conduct a correlation analysis to see the relationships between nutrient contents and enzyme activities. Instead of presenting correlations in the two soils separately as shown in Fig 1, all data should be analyzed together because non-rhizosphere and rhizosphere are just two levels of the same factor (soil type/plant cultivation). Non-rhizosphere should be treated as the control for comparison. By adding an ANOVA table and a correlation table, the authors would be able to see whether plant cultivation impacts a certain nutrient or enzyme activity or not and whether there is any interesting relationship between nutrient content and enzyme activity.
Authors response: Thank you for the suggestion but the one-way ANOVA showed us similar results as the post-hoc, but the principal component analysis of all the data showed interesting relationships between the total soil nutrient concentrations and enzyme activities. Hence, we have included this as figure 2.
Figure 2: correlation between the soil nutrients and associated enzyme activties of E. natalensis rhizosphere and surrounding soils. Soil characteristics are represented as follows: A= nitrate reductase (µmolh-1g-1), B= nitrogen concentration (mg.kg-1), C= acid phosphatase (nmolh-1g-1), D= phosphorus concentration (mg.kg-1), E= β-(D)-Glucosaminidase (nmolh-1g-1), F= alkaline phosphatase (nmolh-1g-1). Principal component analysis (PCA).
Reviewer comment:
In this study, considerable efforts have been made to identify the N-cycling microbes and P-solubilizing microbes in the roots, rhizosphere, and non-rhizosphere; however, the data presentation is quite descriptive and lacks meaningful insights. Although <10% (or even <1%) of all microbes are culturable in the lab, comparing the unique and shared microbes between rhizosphere and non-rhizosphere should still be able to give us at least some insights about who are and who are not there and who might be key players in N, P cycling. To do this, the authors can consider adding a three-way Venn Diagram to see what microbes are unique and what microbes are shared.
Authors response: We have included the three-way Venn Diagram as figure 1 in the manuscript and also, we have quantified the identified microbes according to their functional groups in the results section.
Figure 1: venn diagram of bacteria shared between the rhizosphere and non-rhizosphere soils of E. natalensis.
Reviewer comment:
I am not sure whether the authors did sequence for the rhizosphere soil and non-rhizosphere soil, respectively, as well. The soil sequence data would allow the authors to compare the “population” of the microbes between the rhizosphere soil and the non-rhizosphere soil. Even though the two soils share the same microbial “species”, the “populations” in the two soils could still have a significant difference.
Authors response: Yes, we extracted, sequenced and identified the bacteria in the rhizosphere and non-rhizosphere control soils. The suggested three-way Venn diagram clearly shows the population differences between two soils.
Reviewer comment:
Please see more of my comments below:
- In Table 1, please clarify what Nitrogen content it is. Is it NH4-N, NO3-N, inorganic N, or total N?
Authors response: In this current study we quantified total soil nutrients, including total N and this has been amended in Table 1.
Reviewer comment: 2. Lines 177-178, 180-181: the statements are not correct/precise because the p values in Figs 1A, 1B, 1E, and 1F are >0.05.
Authors response: The statements have been corrected by the authors as suggested by the reviewer.
Reviewer comment: 3. Lines 188-198: keep the unit formats consistent. Either use “-1” or “/” but not both.
Authors response: Thank you for noticing this, to keep the units’ format consistent, we used -1 throughout the manuscript.
Reviewer comment: 4. Discussion Lines 200-213: keep the font, size, and color consistent.
Authors response: We have carefully looked at the manuscript and the font size and color is kept consistent.
Reviewer comment: 5. Author's name is needed for some citations in the text. For example, in line 216 “[46] and [47] described …..” should be replaced with “ Adeniyan et al. [46] and Akinrinlola et al. [47] described….” Double-check lines 255, 258, 262, 267, 269, 274, 277, and 288.
Authors response: We have added the author names in all the in-text citations throughout the manuscript.
Reviewer comment: 6. Lines 300-301 “Soil samples were collected from 0-10 and 10-20 cm depths ….”. Please clarify what soil (0-10 cm, 10-20 cm, or both) was used for soil nutrient analyses and enzyme activity assays.
Authors response: The 0-10 and 10-20 cm depth is considered the portion of soil in closer contact with roots and where maximum microbial activity is expected. The collected soil in each point (10 sub-points) was transferred to a bucket and thoroughly mixed. In total 10 compound samples were collected per site. A portion of each compound soil sample was stored in sterile plastic bags in a refrigerator at 4 °C until chemical and biological analyses were conducted.
Reviewer comment: 7. In 4.1.2, the methods and references for soil nutrient analysis are missing. Please clarify those.
Authors response: A short description of the applied soil nutrient analysis methods have been included in the methods section and reads as follows:
The soil samples were air-dried, sieved to less than 2 mm, and 50 g of each with five replicates were sent for P, N, K, pH, acidity exchange and total cation analysis at the KwaZulu-Natal Department of Agriculture and Rural De-velopment’s Analytical Services Unit, Cedara, South Africa. Ground soil samples were analyzed for total N with the Automated Dumas dry combustion method using a LECO CNS 2000 (Leco Corporation, Michigan, USA) and pH (using a KCl solution). Phosphorus and K in the soil samples were measured using atomic absorption method. This involved the extraction of 2.5 mL soil solution with 25 mL ambic-2 solution at pH of 8. Refer to Manson and Roberts (2000) for the detailed methodologies.
Reference:
Reference: Manson, A.D., Roberts, V.G., 2000. Analytical Methods Used by the Soil Fertility and Analytical Services Section. Republic of South Africa, Pietermaritzburg.
Reviewer comment: 8. In 4.1.3 and 4.1.4, the DNA extraction method and reference are missing. Please clarify those.
Authors response: The polymerase chain reaction (PCR) using small portions of the 16S rDNA genes was done using pure bacterial colonies extracted from the coralloid roots, rhizosphere and non-rhizosphere soils not DNA as per Magadlela et al [37], hence, we have removed DNA extraction in the manuscript.
Reviewer comment: 9. In 4.1.3, the reference for the agars is missing. Please clarify that.
Authors response: The reference for the agars has been included in the relevant sections of the materials and methods.
Authors comment: Thank you for the comments and suggestions, they have certainly improved the manuscript and we are hopeful that the manuscript is now in better shape to be accepted for publication in Plants.

Reviewer 2 Report
Interesting theme and I have some questions about the details (and of course about the interpretation). The goals of the experimental work were:
“is the survival of the South African E. natalensis cycad in nutrient-deficient ecosystem soils a result of its ability to form symbiotic associations with bacteria in its coralloid roots?”
“We hypothesized that the bacteria in the coralloid roots, rhizosphere, and non-rhizosphere soils of the South African E. natalensis cycad contribute to soil nutrient inputs and E. natalensis persistence”
In my opinion, the manuscript does not fulfill its goals. The possibilities of bacterial contribution were justified, but the contribution itself was not supported by proper characterization of rhizosphere soil and enzyme activities.
The experimental data provide the opportunity to achieve the goals, a more thorough discussion required.
Detailed opinion:
Materials and Methods.
Some important things are missing from here, which are needed to the correct explanation of the results.
4.1.1.:
Missing the description of soil type, soil texture.
The data of sampling is missing, and I couldn’t find the number of samples.
4.1.2.:
Not enough to refer the laboratory (who performed the measurement), the naming or short description of the applied methods are important.
It is not always clear whether the reported values refer to the total or available nutrient contents. If they are available concentrations, it must be described according to which method.
4.1.3.:
In case of mentioned methods for bacteria isolation is useful to refer the original method. It would add a lot to the results and their interpretation if the number of N-fixing, N-cycling and P-solubilizing microbes in 1 gram soil was given. If measured.
4.1.6.
In case of the phosphatase measurements, it would be particularly important to specify the pH of buffers.
And it should be explained somewhere in the text, why alkaline phosphatase activity measured in the given acidic soil.
Results.
line 114-115. Rhizosphere soil pH usually lower than the bulk soil (usually the opposite happens). However, the exchange acidity was higher than the non-rhizosphere soil. This contradiction should be explained. In Table 1: what is the difference between the total N and organic-N? The description of measuring method is missing in the Material and Methods.
In addition to the description of the isolated bacteria it would also be interesting to describe how many colonies of the different physiological types (N-fixing, N-cycling, P-solubilizing) were cultured from 1 gram soil.
Figure 1 contain the R values, but using R-squared is more useful, because R-squared has statistical meaning. R-squared (R2) is a statistical measure that represents the proportion of the variance for a dependent variable that's explained by an independent variable or variables in a regression mode.
Discussion
The discussion of the changes in the composition of the rhizosphere soil (comparing to the bulk soil) is missing in this chapter. This is an important question and the correct explanation would also help a lot in the interpretation of enzyme activities.
line 200-212: This general text would fit better in the Introduction.
The description of the properties of isolated bacteria is correct and informative.
line 240-251: This general text would fit better in the Introduction.
In the discussing of phosphatase activities the organic P fraction is mentioned. But this is not mentioned anywhere in the manuscript before, and the measured data does not indicate it either. Usually, the phosphatase activities are lower in the presence of phosphate ions. This is why it would be important to know what P fraction was measured.
The discussion of glucosaminidase activities is very short. The statement in line 279-280 is not justified. Otherwise, it is important to note that nitrate reductase activity reduces the soil N content, since the end product of this process is N2 gas.
mistyping: line 78, Table 1: pH
Author Response
Reviewer 2:
Interesting theme and I have some questions about the details (and of course about the interpretation). The goals of the experimental work were:
Reviewer comment: “is the survival of the South African E. natalensis cycad in nutrient-deficient ecosystem soils a result of its ability to form symbiotic associations with bacteria in its coralloid roots?”
Authors response: No, not solely on their ability to form symbiotic associations with bacteria in their coralloid roots but cycads association with bacteria contributes to their ability to persist in nutrient-poor and harsh environmental conditions. Therefore, we have corrected our statement and currently it reads as follows: Cycad species grow in grasslands, sand dunes, rocky outcrops, scarp and sclerophyll forests, and areas with recurrent fires [5], and their ability to form symbioses with N-fixing, N cycling and P solubilizing bacteria is likely to contributes to their ability to grow and thrive in nutrient-poor and harsh environmental conditions [5,6].
Reviewer comment: “We hypothesized that the bacteria in the coralloid roots, rhizosphere, and non-rhizosphere soils of the South African E. natalensis cycad contribute to soil nutrient inputs and E. natalensis persistence”
In my opinion, the manuscript does not fulfil its goals. The possibilities of bacterial contribution were justified, but the contribution itself was not supported by proper characterization of rhizosphere soil and enzyme activities.
Authors response: We agree with your opinion, hence, we have rephrased our aims and hypothesis and now reads as follows:
The aim of this research was (1) to identify soil bacteria in the coralloid roots, rhizosphere and non-rhizosphere control soil of E. natalensis (2) to correlate to soil nutrition with the enzyme activities of the soil bacteria from the coralloid roots, rhizosphere and non-rhizosphere control soil of E. natalensis. We hypothesized that the bacteria composition in the rhizosphere of coralloid roots of the South African E. natalensis cycad differs from that of the non-rhizosphere control soils of E. natalensis. Increased enzyme activities in the rhizosphere contribute to enhance soil bioavailability that prompts E. natalensis persistence in the nutrient-stressed and disturbed savanna woodland ecosystem soils.
Reviewer comment: The experimental data provide the opportunity to achieve the goals, a more thorough discussion required.
Authors response: We have rewritten and added more aspects in the discussion to achieve the goals of the experimental data.
Detailed opinion:
Materials and Methods.
Some important things are missing from here, which are needed to the correct explanation of the results.
Reviewer comment: 4.1.1.: Missing the description of soil type, soil texture.
Authors response: Some details of soil type and texture has been included in the manuscript.
Reviewer comment: The data of sampling is missing, and I couldn’t find the number of samples.
Authors response: Detailed soil sampling data has been included in the materials and methods section of the manuscript
Reviewer comment: 4.1.2.: Not enough to refer the laboratory (who performed the measurement), the naming or short description of the applied methods are important.
Authors response: A short description of the applied soil nutrient analysis methods have been included in the methods section and reads as follows:
The soil samples were air-dried, sieved to less than 2 mm, and 50 g of each with five replicates were sent for P, N, K, pH, acidity exchange and total cation analysis at the KwaZulu-Natal Department of Agriculture and Rural Development’s Analytical Services Unit, Cedara, South Africa. Ground soil samples were analyzed for total N with the Automated Dumas dry combustion method using a LECO CNS 2000 (Leco Corporation, Michigan, USA) and pH (using a KCl solution). Phosphorus and K in the soil samples were measured using atomic absorption method. This involved the extraction of 2.5 mL soil solution with 25 mL ambic-2 solution at pH of 8. Refer to Manson and Roberts (2000) for the detailed methodologies.
Reference:
Reference: Manson, A.D., Roberts, V.G., 2000. Analytical Methods Used by the Soil Fertility and Analytical Services Section. Republic of South Africa, Pietermaritzburg.
Reviewer comment: It is not always clear whether the reported values refer to the total or available nutrient contents. If they are available concentrations, it must be described according to which method.
Authors response: In this study we analysed the total soil nutrients and this has been highlighted throughout the manuscript
Reviewer comment: 4.1.3.: In case of mentioned methods for bacteria isolation is useful to refer the original method. It would add a lot to the results and their interpretation if the number of N-fixing, N-cycling and P-solubilizing microbes in 1 gram soil was given. If measured.
Authors response: The original methods are included in the materials and methods section. The number of microbes in 1 gram was not measured but the microbes are quantified according to their functions, however, this will be considered in future studies.
Reviewer comment: 4.1.6. In case of the phosphatase measurements, it would be particularly important to specify the pH of buffers.
Authors response: The pH of the buffers is included in the methods section.
Reviewer comment: And it should be explained somewhere in the text, why alkaline phosphatase activity measured in the given acidic soil.
Authors response: In the manuscript, we mention that cattle were grazing in the adjacent grasslands of the E. natalensis population in Edendale resulting to manure deposits in soils. Literature shows an inconsistent relationship between manure and the soil pH. The literature is replete with works that show an increase in pH as a function manure application (Ano and Ubochi, 2007). Consistent increase in soil pH has been reported with the application of rabbit, swine, goat, chicken, and cow manures (Ano and Ubochi, 2007). The increase in the pH as a function of manure application has been attributed to the calcium carbonate and bicarbonate found in manure (Ano and Ubochi, 2007). Due to the above-mentioned, we decided to assay alkaline phosphatase activities in the experimental soils in this current study as the manure may have had effects on the soil pH affecting phosphatase activities.
Reference: Ano, A.O.; Ubochi, C.I. Neutralization of soil acidity by animal manure: Mechanism of reaction. Afr. J. Biotechnol. 2007, 364–368.
Results.
Reviewer comment: line 114-115. Rhizosphere soil pH usually lower than the bulk soil (usually the opposite happens). However, the exchange acidity was higher than the non-rhizosphere soil. This contradiction should be explained. In Table 1: what is the difference between the total N and organic-N? The description of measuring method is missing in the Material and Methods.
Authors response: The contradiction has been explained in the discussion section of the manuscript and the soil analysis methods have been included in the methods section.
Reviewer comment: In addition to the description of the isolated bacteria it would also be interesting to describe how many colonies of the different physiological types (N-fixing, N-cycling, P-solubilizing) were cultured from 1 gram soil.
Authors response: Unfortunately, the colonies per 1gram soil were not counted in this current study as our main aim was to identify the bacteria forming the colonies and quantify the bacteria using their functions but this will be considered in future studies. Thank you.
Reviewer comment: Figure 1 contain the R values, but using R-squared is more useful, because R-squared has statistical meaning. R-squared (R2) is a statistical measure that represents the proportion of the variance for a dependent variable that's explained by an independent variable or variables in a regression mode.
Authors response: We have excluded this data from the manuscript and included a principal component analysis of all the data to establish the relationship between the assayed total nutrient concentrations and enzyme activities as suggested by reviewer 1.
Discussion
Reviewer comment: The discussion of the changes in the composition of the rhizosphere soil (comparing to the bulk soil) is missing in this chapter. This is an important question and the correct explanation would also help a lot in the interpretation of enzyme activities.
Authors response: A section discussing the experimental soils is included in the discussion.
Reviewer comment: line 200-212: This general text would fit better in the Introduction.
Authors response: The text has been removed from the discussion as suggested by reviewer.
Reviewer comment: The description of the properties of isolated bacteria is correct and informative.
Authors response: Thank you
Reviewer comment: line 240-251: This general text would fit better in the Introduction.
Authors response: The text has been removed from the discussion as suggested by reviewer.
Reviewer comment: In the discussing of phosphatase activities the organic P fraction is mentioned. But this is not mentioned anywhere in the manuscript before, and the measured data does not indicate it either. Usually, the phosphatase activities are lower in the presence of phosphate ions. This is why it would be important to know what P fraction was measured.
Authors response: In the current study we measure total soil P concentrations not individual P fractions and this has been detailed in the manuscript.
Reviewer comment: The discussion of glucosaminidase activities is very short.
Authors response: We have added more discussion related to glucosaminidase activities.
Reviewer comment: The statement in line 279-280 is not justified. Otherwise, it is important to note that nitrate reductase activity reduces the soil N content, since the end product of this process is N2 gas.
Authors response: Nitrate reduction to ammonium (NRA) is one of several important processes that are responsible for nitrogen cycling in soil. Both bacteria and fungi have been found to be capable of carrying out NRA (Takaya, 2002; Philippot, 2005) and the nrfA gene has been shown to be present in a wide variety of bacteria (Smith et al., 2007). Due to these studies, we concluded it was important to assay nitrogen reductase in the current experimental soils and correlate the nitrogen reductase activities with total soil nitrogen concentrations.
References:
Philippot, L. (2005). Tracking nitrate reducers and denitrifiers in the environment. Biochem. Soc. Trans. 33, 200–204.
Takaya, N. (2002). Dissimilatory nitrate reduction metabolisms and their control in fungi. J. Biosci. Bioeng. 94, 506–510.
Smith, C. J., Nedwell, D. B., Dong, L. F., and Osborn, A. M. (2007). Diversity and abundance of nitrate reductase genes (narG and napA), nitrite reduc-tase genes (nirS and nrfA), and their transcripts in estuarine sediments. Appl. Environ. Microbiol. 73, 3612– 3622.
Reviewer comment: mistyping: line 78, Table 1: pH
Authors response: The pH mistyping has been corrected. Thanks
Authors comment: Thank you for the comments and suggestions, they have certainly improved the manuscript and we are hopeful that the manuscript is now in better shape to be accepted for publication in Plants.

Round 2
Reviewer 1 Report
The authors did a great job in revising the manuscript. I don't have further comments.
Author Response
Reviewer 1: The authors did a great job in revising the manuscript. I don't have further comments.
Author comments: Thank you
Reviewer 2 Report
I accept corrections and answers to my questions (with one exception, later in this text). Thank you for considering my comments. The manuscript is better than the previous version in many aspects, and provides new information about the cycads, their microbial partners and their role. In general, I consider it worthy of publication in the journal Planta.
However, one part of the manuscript is still unsatisfactory: this is the description of the characteristics of the soils, the concentrations of nutrients. Obviously, this is not the central part of the research topic, but these soil data are essential for the evaluation of enzyme activities and nutrient supply.
According to reference 119 (Manson and Roberts, 2000), the values of P, K, Mg, Ca, Zn, Mn and Cu are not total, but available concentrations: "Ambic-2-extractable P, K, Cu, Mn and Zn, KCl-extractable Ca, Mg" (quote from reference 119). The descriptions in Table 1 and elsewhere in the text should be modified accordingly to this. This is an important difference and explains many things in the manuscript. The available concentrations of nutrients can be modified by soil enzymes much more than the total amounts. The available concentrations of nutrients are usually higher in the rhizosphere than in the bulk soil. This is the result of increased microbial activity, and this is in accordance with the published results.
Another strange thing in Table 1, that the amount of organic N (0,445% = 4450 mg/kg) is greater than that of total N (4187 mg/kg) in rhizosphere. Which is obviously not possible. The organic N measurement is missing (only the total N measurement is described), so the reason is unknown.
After correcting these soil data, I recommend the manuscript to publish.
Best regards,
Author Response
Reviewer 2: I accept corrections and answers to my questions (with one exception, later in this text). Thank you for considering my comments. The manuscript is better than the previous version in many aspects, and provides new information about the cycads, their microbial partners and their role. In general, I consider it worthy of publication in the journal Plants.
Author comments: Thank you
Reviewer 2: However, one part of the manuscript is still unsatisfactory: this is the description of the characteristics of the soils, the concentrations of nutrients. Obviously, this is not the central part of the research topic, but these soil data are essential for the evaluation of enzyme activities and nutrient supply.
According to reference 119 (Manson and Roberts, 2000), the values of P, K, Mg, Ca, Zn, Mn and Cu are not total, but available concentrations: "Ambic-2-extractable P, K, Cu, Mn and Zn, KCl-extractable Ca, Mg" (quote from reference 119). The descriptions in Table 1 and elsewhere in the text should be modified accordingly to this. This is an important difference and explains many things in the manuscript. The available concentrations of nutrients can be modified by soil enzymes much more than the total amounts. The available concentrations of nutrients are usually higher in the rhizosphere than in the bulk soil. This is the result of increased microbial activity, and this is in accordance with the published results.
Author comments: Thank you for the comments. Yes, we agree that the analysed soil P, K, Cu, Mn and Zn concentrations are Ambic-2-extractable and the soil Ca, Mg concentrations are KCl-extractable and Total C, N and Sulphur using the Automated Dumas dry combustion method using a LECO CNS 2000. This has been corrected in Table 1 and materials and methods sections and other sections of the manuscript.
Reviewer 2: Another strange thing in Table 1, that the amount of organic N (0,445% = 4450 mg/kg) is greater than that of total N (4187 mg/kg) in rhizosphere. Which is obviously not possible. The organic N measurement is missing (only the total N measurement is described), so the reason is unknown.
Author comments: Thank you and we agree with you, therefore, we have decided to exclude % organic N data and solely present total N data, which was calculated using the Automated Dumas dry combustion method using a LECO CNS 2000 and this method is included and referenced in the materials and methods in the manuscript. Most importantly, the total N data was the N data used in the Principal component analysis (PCA) to correlate soil nutrition with the enzyme activities as per the second aim of the study.
Reviewer 2: After correcting these soil data, I recommend the manuscript to publish.
Best regards,
Author comments: Thank you once again as the suggested correction improved the manuscript.